# Enantioselective synthesis of chiral amides by carbene insertion into amide N−H bond

Xuan-Ge Zhang[1], Zhi-Chun Yang[1], Jia-Bin Pan [1], Xiao-Hua Liu [2] & Qi-Lin Zhou [1] ✉

Chiral amides are important structure in many natural products and pharmaceuticals, yet their efficient synthesis from simple amide feedstock remains challenge due to its weak Lewis basicity. Herein, we describe our study of the enantioselective synthesis of chiral amides by N-alkylation of primary amides taking advantage of an achiral rhodium and chiral squaramide co-catalyzed carbene N−H insertion reaction. This method features mild condition, rapid reaction rate (in all cases 1 min) and a wide substrate scope with high yield and excellent enantioselectivity. Further product transformations show the synthetic potential of this reaction. Mechanistic studies reveal that the non-covalent interactions between the catalyst and reaction intermediate play a critical role in enantiocontrol.

Chiral amides are common structural units in natural products, pharmaceuticals, and agrochemicals (Fig. 1a)[1–5], therefore, their efficient synthesis is of great significance and has received increasing attention. Traditional synthesis of chiral amides is the condensation of carboxylic acids or their derivatives with chiral amines[6], which has been extensively studied and relies highly on coupling agents to facilitate dehydration. As a complementary approach, the assemble of chiral amide through asymmetric N-alkylation of readily available primary amide feedstock provides an opportunity to create new chiral center in a catalytic manner (Fig. 1b). However, due to the weak nucleophilicity of amide caused by the delocalization of their N-lone pair into the carbonyl group as well as the difficulty in controlling stereochemistry during the C($sp^3$)−N bond formation step, practical method for highly enantioselective N-alkylation of amide remains rare with only a few successful examples using secondary alkyl halide[7], benzylic C−H bond[8,9], imine[10,11], as alkylation agent. Thus, the development of alternative amide N-alkylation protocols that can efficiently produce chiral amide under mild condition is highly desired. In this context, the enantioselective carbene N−H bond insertion[12–14], featuring high reactivity, mild reaction conditions and good functional group tolerance[15,16], provides a suitable technique for N-alkylation of amide.

In recent years, asymmetric N−H bond insertion reactions have been applied successfully to various types of amines, such as aromatic amines[17–23], carbamates[24–28], aliphatic amines[29,30], and ammonia[31], with high yield and excellent enantioselectivity. However, to the best of our knowledge, highly enantioselective N−H insertion of amides remains a challenge, with only one example of enantioselective N−H insertion reaction between diazoesters and amides reported, with ee values up to 77%[32]. This may be a function of properties specific to amides. First, the low Lewis basicity of amides interferes with its bonding to metal-carbene species. Second, the ylide intermediate that forms from the nucleophilic addition of the amide to the metal carbene has a comparatively high acidity, favoring a tendency toward deprotonation rather than interaction with a chiral catalyst (Fig. 2a).

To overcome these challenges, we need to find a chiral organocatalyst that could efficiently mediate the proton transfer process of the reaction intermediates. Through a systematic screen of catalysts, we found that chiral squaramides with a quinine skeleton can effectively control the enantioselectivity of the amide N−H bond insertion reaction of carbenes generated from diazoketone decomposition. Herein, we describe our strategy for enantioselective N-alkylation of primary amide using carbene insertion into amide N−H bond (Fig. 2b). Under mild condition, low catalyst loading (1 mol% for both metal and organocatalyst) and fast reaction rate (1 min in all cases), the reaction efficiently installs a sp3-C chiral center to the amide feedstock. This method provides a powerful tool for the synthesis of structural divergent chiral amides and related useful molecules (see General Information in Supplementary Information for details).

[1]State Key Laboratory and Institute of Elemento-Organic Chemistry, College of Chemistry, Frontiers Science Center for New Organic Matter, Nankai University, Tianjin, China. [2]Key Laboratory of Green Chemistry & Technology, Ministry of Education, College of Chemistry, Sichuan University, Chengdu, China. ✉e-mail: qlzhou@nankai.edu.cn

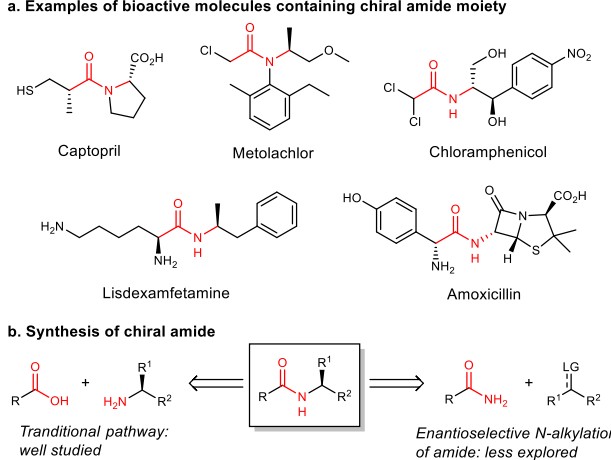

a. Examples of bioactive molecules containing chiral amide moiety

Captopril

Metolachlor

Chloramphenicol

Lisdexamfetamine

Amoxicillin

b. Synthesis of chiral amide

Tranditional pathway: well studied

Enantioselective N-alkylation of amide: less explored

**Fig. 1 | Synthesis of chiral amide. a** Exhibition of bioactive molecules that conation chiral amide structure. **b** Strategies of chiral amide synthesis: traditional pathway and our way.

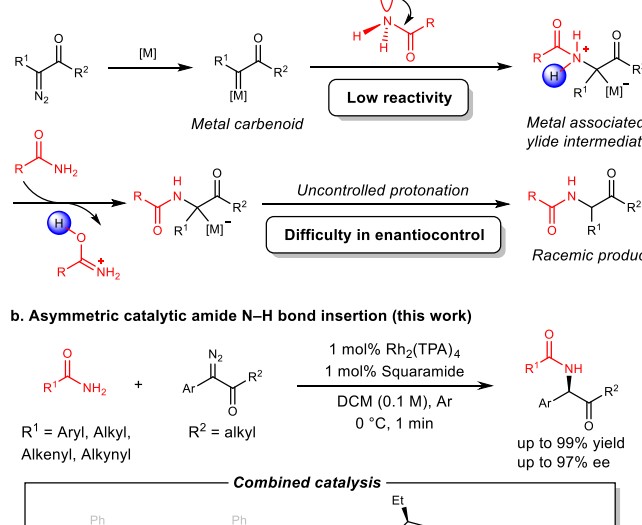

a. Challenges of enantioselective N–H bond insertion using primary amide

Metal carbenoid

Low reactivity

Metal associated ylide intermediate

Uncontrolled protonation

Difficulty in enantiocontrol

Racemic product

b. Asymmetric catalytic amide N–H bond insertion (this work)

R¹ = Aryl, Alkyl, Alkenyl, Alkynyl

R² = alkyl

1 mol% Rh₂(TPA)₄
1 mol% Squaramide

DCM (0.1 M), Ar
0 °C, 1 min

up to 99% yield
up to 97% ee

*Combined catalysis*

Rh₂(TPA)₄

Squaramide

**Fig. 2 | Enantioselective carbene insertion of amide N–H bond. a** Difficulties in enantioselective amide N–H bond insertion reaction. **b** This work: Rhodium triphenylacetate and chiral squaramide co-catalyzed asymmetric amide N–H insertion.

## Results

### Optimization of reaction conditions

We began by exploring reaction conditions using 3-phenylpropionamide (**1a**) and 1-diazo-1-phenylpropan-2-one (**2a**) as model substrates (Table 1; see Reaction Optimization section in Supplementary Information for details). When copper catalysts such as Cu(MeCN)$_4$PF$_6$ and CuTp* (copper hydrotris(3,5-dimethylpyrazolyl) borate) were used to decompose the diazoketone, no desired product was detected (entries 1, 2). The use of dirhodium(II) acetate (Rh$_2$(OAc)$_4$) afforded the insertion product **3a** with 80% yield (entry 3). We then used Rh$_2$(OAc)$_4$ in combination with organocatalysts to control the enantioselectivity of the reaction. Chiral spiro phosphoric acids and amides **C1**–**C3**, which have been demonstrated to be enantioselective proton transfer catalysts in the previously reported carbamate N–H insertion reactions[33], gave only very low ee in this reaction (<10%) (entries 4–6). Organocatalysts **C4**–**C7** of urea, thiourea, and squaramide with chiral amino groups showed low to moderate enantioselectivity (entry 7–10). To our delight, introduction of a hydroquinine or hydroquinidine skeleton to the squaramide catalyst (**C8** and **C9**) significantly improved the enantioselectivity to 80% ee (entry 11, 12). Diastereomeric catalysts **C8** and **C9** yielded enantiomers of the insertion product with the same ee values, which offers a choice for obtaining chiral amides with different configurations. Further evaluation of dirhodium catalysts showed that substitution on the carboxylic acid ligand impacts the enantioselectivity of the reaction (entry 13–16), with triphenylacetic acid being the best ligand (entry16, Rh$_2$(TPA)$_4$, 88% ee). Furthermore, the use of Rh$_2$(TPA)$_4$ significantly accelerated the reaction, which was completed within 1 min, and increased the enantioselectivity of the reaction to 90% ee (entry 17). Adding the diazoketone dropwise and lowering the reaction temperature to 0 °C further improved the yield and enantioselectivity (entries 18 and 19).

### Substrate scope

Under the optimal reaction conditions, the scope of substrates was studied (Fig. 3; see Synthesis and Analytical Data in Supplementary Information for details). Surprisingly, reactions using substrates with all types of structure and functional groups were finished within 1 min after the setup, showing the efficiency of this method. First, a broad range of amides were investigated in the reaction with diazoketone **2a** (Fig. 3a). Primary amides with different chain lengths underwent the reaction smoothly to afford the corresponding α-amido ketones (**3a**–**3e**) with high yield (96–99%) and excellent enantioselectivities (90–93% ee). Amides with bulkier alkyl group gave higher

enantioselectivity but lower yield (**3f** and **3g** vs **3d**). Amides with cycloalkyl or saturated heterocycles afforded insertion products (**3h**–**3o**) in high yields (85–99%) and with high enantioselectivities (88–96% ee). In addition to aliphatic amides, aromatic amides bearing either an electron-donating substituent (**3q**), an electron-withdrawing substituent (**3r**), or a halogen (**3s**–**3v**) all gave desired products with high yields (93–99%) and high enantioselectivites (92–96% ee).The reaction using amides having naphthyl, furyl, thiophenyl, pyridyl, and indole groups also underwent well, affording the desired products (**3w**–**3aa**) in high yields with high enantioselectivities. It is worth noting that since the basicity of the quinuclidine group of catalyst **C8** is much higher than that of pyridyl, the pyridyl group will not interfere with the reaction. Moreover, for 1H-indole-2-carboxamide, the insertion reaction selectively occurs at amide N–H rather than at indole N–H. Acrylamide and propynamide were also suitable substrates for the reaction, affording insertion products (**3ab** and **3ac**) in satisfactory yield and ee. These results demonstrated that the amide insertion reaction has good functional group tolerance.

Next, the substrate scope with respect to the diazoketones was investigated in the amide N–H insertion reaction (Fig. 3b). Installation of electron-donating substituents (**4a, 4b, 4m**) or electron-withdrawing substituents (**4c**–**4i, 4o**–**4q**) and phenyl (**4k**) on the aryl ring of the diazoketones resulted in good yields and moderate to high enantioselectivities (74–93% ee). The halogen atoms in the products offer opportunities for subsequent synthetic transformations. The diazoketones containing a naphthyl group (**4ij** and **4n**) also worked well and gave high enantioselectivities (94% ee and 91% ee, respectively). To our delight, the switch of methyl to ethyl of the diazoketone substrate had no significant effect on the enantioselectivity of the reaction (**4l**). We then used this protocol to investigate the derivatizations of various pharmaceutical agents and other bioactive molecules, such as indomethacin, rufinamide, oxaprozin, naproxen, levetiracetam, penicillin, glutamine, and asparagine (Fig. 3c). Under the standard conditions, these compounds smoothly underwent the asymmetric insertion reaction to afford the corresponding chiral

**Table 1 | Reaction optimization[a]**

| entry | [M] | catal. | yield (%)[b] | ee (%)[c] |
|---|---|---|---|---|
| 1 | Cu(MeCN)$_4$PF$_6$ | none | ND | none |
| 2 | CuTp* | none | ND | none |
| 3 | Rh$_2$(OAc)$_4$ | none | 80 | none |
| 4 | Rh$_2$(OAc)$_4$ | C1 | 82 | 8 |
| 5 | Rh$_2$(OAc)$_4$ | C2 | 80 | 0 |
| 6 | Rh$_2$(OAc)$_4$ | C3 | 83 | 2 |
| 7 | Rh$_2$(OAc)$_4$ | C4 | 84 | 15 |
| 8 | Rh$_2$(OAc)$_4$ | C5 | 85 | 21 |
| 9 | Rh$_2$(OAc)$_4$ | C6 | 82 | 52 |
| 10 | Rh$_2$(OAc)$_4$ | C7 | 79 | 33 |
| 11 | Rh$_2$(OAc)$_4$ | C8 | 80 | 80 |
| 12 | Rh$_2$(OAc)$_4$ | C9 | 82 | 80 (−)[d] |
| 13 | Rh$_2$(oct)$_4$ | C8 | 81 | 81 |
| 14 | Rh$_2$(TFA)$_4$ | C8 | 75 | 71 |
| 15 | Rh$_2$(piv)$_4$ | C8 | 92 | 76 |
| 16 | Rh$_2$(TPA)$_4$ | C8 | 70 | 88 |
| 17[e] | Rh$_2$(TPA)$_4$ | C8 | 70 | 90 |
| 18[e, f] | Rh$_2$(TPA)$_4$ | C8 | 87 | 90 |
| 19[e, f, g] | Rh$_2$(TPA)$_4$ | C8 | 99 | 92 |

[a]Reaction conditions: 3-phenylpropanamide **1a** (0.2 mmol), 1-diazo-1-phenylpropan-2-one **2a** (0.2 mmol), DCM (2 mL), 20 °C, 12 h.
[b]Isolated yields.
[c]The ee values were determined by chiral HPLC.
[d]The configuration is opposite to those obtained by using other catalysts.
[e]Reaction was finished within 1 min.
[f]Diazoketone solution was added dropwise.
[g]Reaction was performed at 0 °C.

N-alkylated amides (**5a**–**5h**) with high yields (82–99%) and good enantioselectivities (83–90% ee) or diastereoselectivities (86:14 to >99:1). Products **5f**–**5h** show that N-alkylation occurs on primary amides but not on secondary amides and carbamates. These results demonstrate the applicability of this protocol for the synthesis of drug-like molecules.

## Applications

To further demonstrate the application potential of the method in synthesis, we carried out scale-up experiments and a series of product transformations. At 1 mmol scale, **3l** and **3p** can be produced smoothly without obvious decrease of yield or ee (Fig. 4a). In the synthesis of chiral compounds, a good method should be able to selectively prepare each enantiomer. To our delight, under standard conditions, catalysts **C8** and **C9**, which are diastereomers, produced the insertion product **3a** and ent-**3a**, respectively, with the same yield and ee but with opposite configuration (Fig. 4b). Given the ubiquity of chiral amino alcohols[34–37], especially tertiary alcohols[38], in natural products, biologically active molecules, and chiral ligands and auxiliaries, we conducted Grignard reactions on the insertion product **3p**, producing

chiral 1,2-amino tertiary alcohols (Fig. 4c; see Determination of the Configuration of **7d** in Supplementary Information for details). The chiral center adjacent to the ketone was not epimerized by the strongly basic Grignard reagent and effectively controlled the newly formed chiral center with excellent diastereoselectivities (dr >20:1 in all cases, **7a**–**7h**). Wittig reaction of **3p** gave allylic amide **8**, and reduction afforded the 1,2-vicinal amino alcohol **9**, which was further transformed into chiral phenyl oxazoline **10** through a MsCl mediated cyclization (Fig. 4d)[39–41]. In addition, the dehydrogenase inhibitor phosphoglycerate **12**[42] was synthesized in excellent 98% yield, 96% ee, and >20:1 diastereoselectivity by amide insertion followed by a reduction (Fig. 4e).

## Mechanism studies

The mechanism of the amide N–H bond insertion reaction was presumed to be similar to that of the hydrogen bonding donor (HBD) catalyzed carbene insertion reaction, which has been extensively studied in our group and others[27,29,30,43,44]. First, dirhodium(II) triphenylacetate **I** reacts with diazaoketone **2a** to generate a highly reactive Rh-carbenoid intermediate **II** by releasing one equivalent N$_2$ (Fig. 5a). The

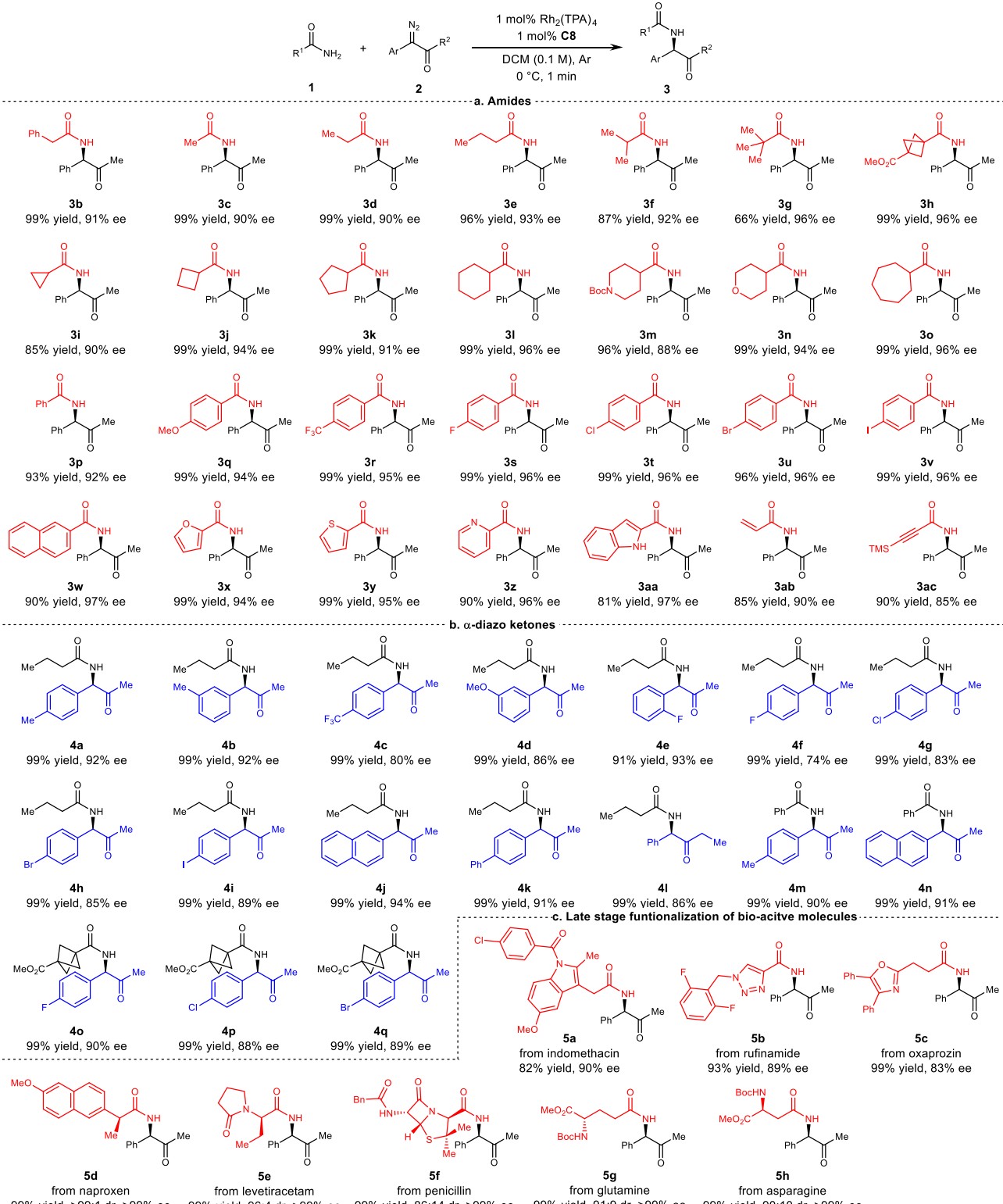

**Fig. 3 | Scope of substrates and late stage functionalization of bioactive molecules.** All insertion reactions were conducted on 0.2 mmol scale. Isolated yields are given. **a** Substrate scope of amide. **b** Substrate scope of α-diazoketone. **c** Late stage functionalization of bioactive molecules.

Rh-carbenoid species undergoes nucleophilic addition with amide **1c** to form a metal associated ylide intermediate **III**, which is rapidly decomposed to give free enol **IV**. The free enol is captured by catalyst **C8** to form a hydrogen bonding complex **V**, and the proton of squaramide is attracted by the electron-negative carbon of the enol to form the intermediate **VI**. Intermediate **VI** undergoes an enantioselective

proton transfer to afford product **3c** and releases the squaramide catalyst **C8**. A plot of log(er) values versus Hammett σ values of the insertion products generated from various substituted diazoketones shows a linear correlation between enantioselectivity and the electronic effect of the substituents (Fig. 5b), which shows that the carbon linked to the substituted phenyl group has a certain amount of charge

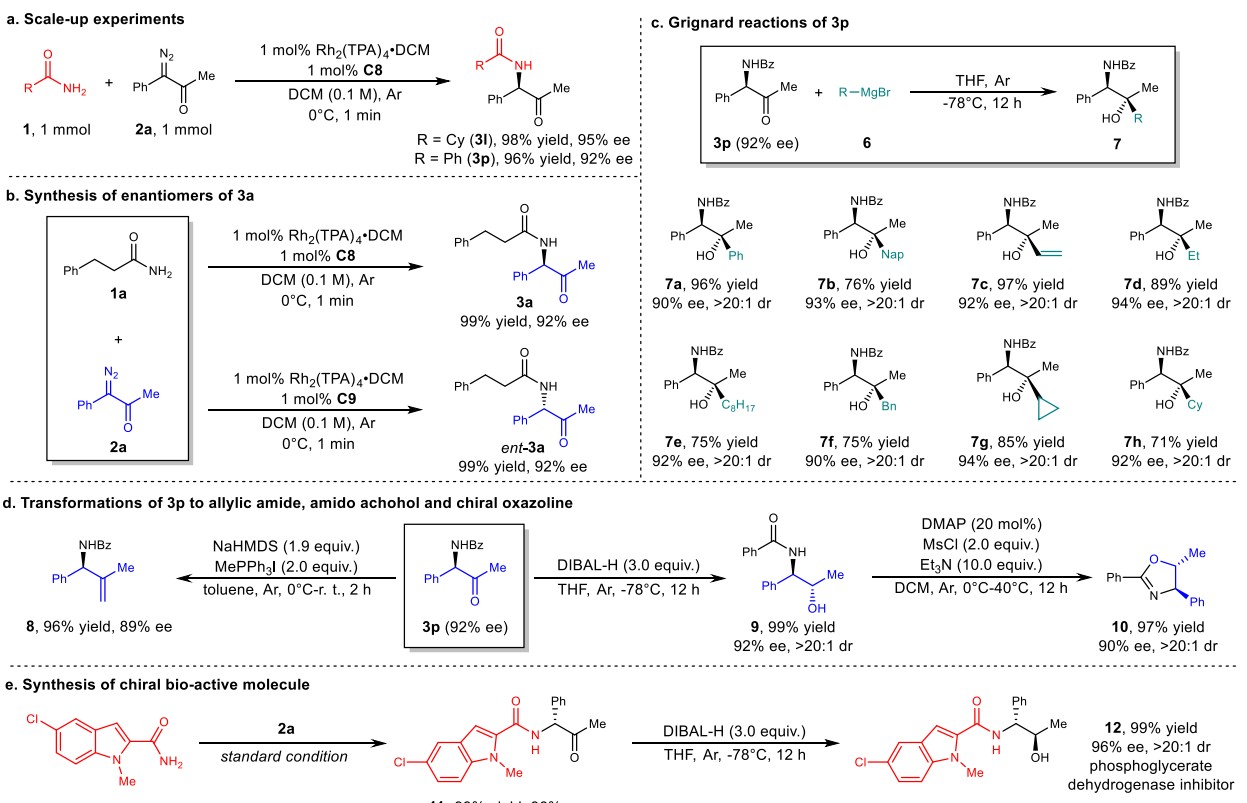

**Fig. 4 | Scale-up experiments and transformations of insertion products.**
**a** Scale-up synthesis. **b** Enantio-divergent synthesis of **3a**. **c** Scope of Grignard
addition reaction of **3p** (all in 0.05 mmol scale). **d** Transformation of **3p** to chiral
allylic amide, amido-alcohol and chiral oxazoline. **e** Enantio- and diastereo-selective
synthesis of phosphoglycerate dehydrogenase inhibitor **12**.

in the transition state. The relatively small slope ($\rho = -0.69$) indicates
that there is no obvious charge separation in the enantio-determining
step and the proton transfer more likely proceeds in a concerted way
through a cyclic TS[45]. Control experiments were also conducted to
examine the role of the carbonyl group of the diazo substrate (Fig. 5c).
The insertion reactions of α-diazoester and diazoamide with butana-
mide yielded the desired products with low ee. Removing the carbonyl
group from the diazo compound led to messy results, and no insertion
product was obtained. These results revealed that an electron-
deficient carbonyl group in the diazo substrate is essential in increas-
ing the electrophilicity of metal-carbene intermediate to gain reactivity
with amide and efficiently bonding with the chiral catalyst.

To further understand the origin of the enantioselectivity of the
reaction, we studied the proton transfer step considering the influ-
ence of rhodium complex using density functional theory calcula-
tions (Fig. 6). Previous works on copper(I) and HBD co-catalyzed
insertion reaction suggest that the coordination of Cu with the sulfur
atom of the thiourea or the oxygen atom of the squaramide plays
important role in lowering the energy barrier of the proton transfer
step[29,31]. However, the same effect was also found in the Rh(II) and
squaramide cooperative catalytic system, and the coordination of
dirhodium with hydroquinine nitrogen atom significantly lowered
the energy barrier by 6.6 kcal/mol (see Computational Study in Sup-
plementary Information and Computational Data in Supplementary
Data 1 for details). Therefore, we analyzed the proton transfer step,
including the participation of the dirhodium catalyst. The lowest-
energy transition state structures for the major and minor enantio-
mers of the product, **RhTPA-TSR** and **RhTPA-TSS**, are presented in
Fig. 6, in which the quinuclidine of catalyst abstracts the proton of
the enol hydroxyl group and the squaramide donates its proton to
the enol carbon. The energy difference between **RhTPA-TSR** and
**RhTPA-TSS** is 2.2 kcal/mol, resulting in the major product with (R)-

configuration, which agrees with the experimental observations.
Both transition states have multiple hydrogen bonding interactions
between the chiral squaramide catalyst and the enol intermediate.
However, more hydrogen bonds and stronger π–π stacking interac-
tion were found in **RhTPA-TSR** than in **RhTPA-TSS** by weak inter-
action analysis of the independent gradient model based on the
Hirschfeld partition (IGMH)[46–49], suggesting that the enantioselec-
tivity was controlled mainly by non-covalent interactions.

## Discussion
In conclusion, we have established an efficient and straightforward
method for the enantioselective N-alkylation of primary amides
through carbene insertion into amide N–H bond. This method features
mild condition, rapid reaction rate and broad substrate scope. The
success of this transformation relies on the combined properties of the
achiral dirhodium catalyst and the chiral squaramide catalyst. DFT
studies and IGMH analysis show that the non-covalent interactions
between the catalyst and reaction intermediate are decisive for the
enantiocontrol. Future efforts will focus on applying this strategy to
the alkylation of amides using other alkylation agents.

## Methods
### General procedure for enantioselective N−H insertion of amide 1a with diazoketone 2a
The Rh$_2$(TPA)$_4$·DCM (2.9 mg, 0.002 mmol, 1 mol%), chiral squaramide
catalyst (1.3 mg, 0.002 mmol, 1 mol%), and 3-phenylpropionamide **1a**
(30.0 mg, 0.2 mmol, 1.0 eq.) were introduced into an oven-dried
Schlenk tube containing a stir bar in a argon-filled glove box. Then, the
Schlenk tube was sealed with a rubber, moved out of the glove box and
injected with 1.5 mL DCM. The tube was cooled down to 0 °C and
0.5 mL of DCM solution of 1-diazo-1-phenylpropan-2-on **2a** (32.0 mg,
0.2 mmol; diazo compound may be thermally unstable and should be

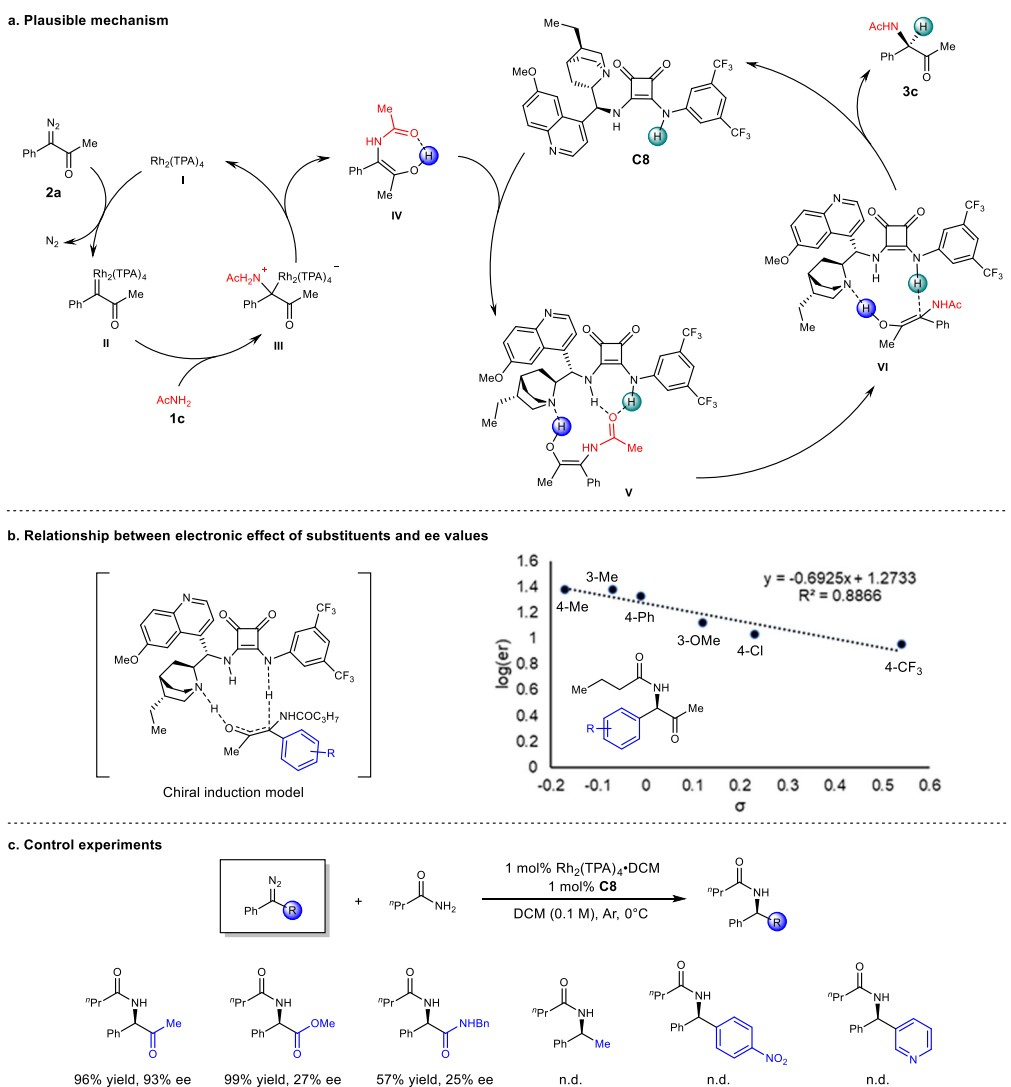

**Fig. 5 | Plausible mechanism and mechanistic investigations. a** Plausible mechanism. **b** Correlation of enantioselectivity (e.r.) with Hammett constant of α-diazoketones. **c** Control experiments showing the importance of the carbonyl group of α-diazoketones.

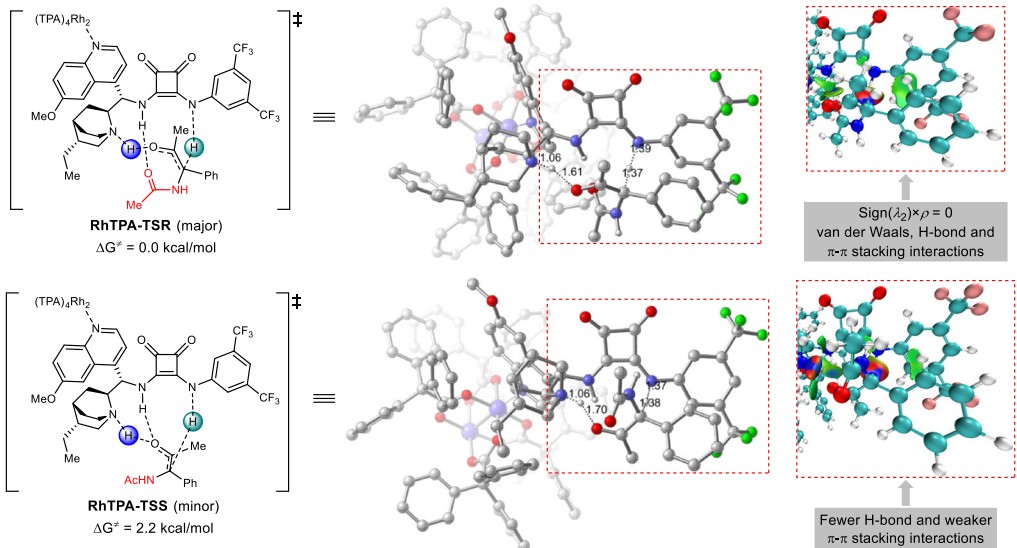

**Fig. 6 | DFT study and IGMH analysis.** Lowest-energy transition structures for *R* and *S* products optimized by DFT calculations performed at the b3lyp-D3(BJ)/def2tzvpp (SMD-dichloromethane)//b3lyp-D3(BJ)/def2svp (gas) level. $\rho$, electron density; sign($\lambda_2$), the sign of the second eigenvalue $\lambda_2$ of the electron density Hessian matrix.

handled carefully with personal protective equipment) was added dropwise within 5 mins via a syringe while stirring. The resulting mixture was allowed to stir for another 1 min at 0 °C. Upon completion, organic volatiles were evaporated in vacuo, and the residue was purified by flash chromatography purification (PE/EA = 3:1, v/v). Product **3a** was obtained as a withe solid (56.5 mg, 99% yield) and characterized by NMR and HPLC (see NMR Spectra and HPLC Spectra in Supplementary Information for details).

## Data availability

All data regarding materials and methods, optimization studies, experimental procedures, DFT calculations, NMR spectra and HPLC spectra can be found in the Supplementary Information. The Cartesian coordinates of the optimized structures are included in the Supplementary Data. All other data are available from the corresponding authors upon request.

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

## Acknowledgements

The authors thank the National Key R&D Program of China (2022YFA1504302, Q.-L.Z.), the National Natural Science Foundation of China (22188101, Q.-L.Z., 91956000, Q.-L.Z., 92256301, Q.-L.Z.), the Fundamental Research Funds for the Central Universities, and the Haihe Laboratory of Sustainable Chemical Transformations for financial support.

## Author contributions

Q.-L.Z. conceived the study; X.-G.Z. and Q.-L.Z. designed the experiments and analyzed the data; X.-G.Z., Z.-C.Y., J.-B.P. and X.-H.L. synthesized substrates and catalysts. X.-G.Z. performed the reactions and mechanistic study. X.-G.Z. and Q.-L.Z. wrote the manuscript.

## Competing interests

The authors declare no competing interests.
