## [Peer Review File · Nature Communications]

Enantioselective Synthesis of Chiral Amides by Carbene Insertion into Amide N–H BondREVIEWER COMMENTS

Reviewer #1 (Remarks to the Author):

The manuscript submitted by Zhou, Liu and coworkers reported achiral rhodium and chiral squaramide co-catalyzed enantioselective carbene N–H insertion reaction, which provides a powerful tool for the synthesis of structural divergent chiral amides and related useful molecules. The authors have optimized the conditions to achieve high enantiomeric excesses for broad substrate scope under mild reaction conditions and at a rapid reaction rate. The reaction products were converted into valuable derivatives. Kinetic studies and computational studies were performed to elucidate the asymmetric proton transfer process in the reaction. Therefore, I would like to recommend the publication of this paper in Nat. Commun. if the authors can revise the manuscript appropriately. I would like to make some comments below.

1. Although rhodium catalysts work well in Table 1 and SI, have authors tried other metal catalysts besides $\text{Cu}(\text{MeCN})_4\text{PF}_6$ and CuTP^* ? I feel that many metal catalysts can catalyze this process, such as Pd, Fe catalysts etc.
2. In DFT studies, the complexation of rhodium catalyst and squaramide(C=O) was ruled out by DFT calculations. Is it possible that chiral Rh complexes could be formed by the coordination interaction of $\text{Rh}_2(\text{TPA})_4$ with nitrogen atom of hydroquinine?
3. It is interesting to know if this enantioselective carbene N–H bond insertion transformation is applicable to the alkyl-substituted diazo compounds.

Reviewer #2 (Remarks to the Author):

Chiral amide bonds are ones of the most common in nature and their efficient synthesis remains challenging. Zhou and his co-workers report an efficient chiral amide synthetic methodology with a pretty rapid reaction rate under mild condition via the N-H insertion. The co-catalyst system works efficiently to afford an excellent enantioselective control. The reaction also gave a wide substrate scope and synthetic application in the chiral amide bond construction. I reaccommodate it to be published after the following questions are addressed:

1. It is seemed that the substrate, diazoketones, play important roles in the enantioselective control. As the authors illustrate in the reaction mechanism, the carbonyl group may serve as a hydrogen bond donor/acceptor in the key intermediates, then what will happen if remove the carbonyl group? and how about phenylacetate pyridine diazo? This should be mentioned and discussed in the mechanism investigation section.
2. The proton shift is key in this reaction, so how water affects this reaction? Does it need dry solvent?
3. Do the chiral Rh catalysts work in this reaction?
4. How about the ee value for each diastereomer of 5e-5h? These data need to be addressed.
5. It is a little confused, the sentence (Line 111-112) "In the synthesis of chiral compounds, it is important to be able to selectively obtain either enantiomer." What the authors would like to declare?
6. Many language errors need to be corrected, for example:
Line 24, carbonyl should be carbonyl group
Line 64, the same ee value should be the same ee values
Line 82, Amide should be amides, alkyl should be alkyl group

Line 84 and 86, with high yields ... should be in high yields and with high enantioselectivities...

Line 87, "The amidesalso underwent the reaction" should be "The reaction using amides.....also underwent well."

Line 88, with high yields should be in high yields

Line 105, showed should be show

Line 106, demonstrated should be demonstrate; "to" should be "for"

Line 120, with excellent yield should be in excellent yield

Line 130, decomposes should be is decomposed

Line 133, release should be releases

Reviewer #3 (Remarks to the Author):

In this manuscript, Zhou et al. report an enantioselective carbene insertion into amide N-H bonds catalyzed by an achiral rhodium complex and a chiral squaramide. The catalyst loadings are low, the reaction times short, and both yields and enantioselectivities high. The scope is good, but more diversity and information about limitations are warranted. DFT is used to explain the enantiocontrol enforced by the squaramide catalyst, but the observed experimental effect of the rhodium catalyst on enantioselectivity is not discussed. Although enantioselective carbene insertion into N-H bonds have been reported before, the amides in this paper are significantly different from previous nitrogen coupling partners. As such, I would evaluate the novelty to be suitable for Nature Communication.

Overall, there are some issues to address; nonetheless, the manuscript could become suitable for Nature Communication after addressing the comments outlined below.

Two places in the introduction, I find the phrasing unfair: "...reduction of hazardous by-products..." and "...N₂ as the sole environmentally friendly by-product". The α -diazo ketone is not a naturally occurring feedstock material; it must be synthesized. As such the α -diazo ketone synthesis should be included in the consideration of the environmental impact, especially when the authors are comparing it to amide bond formation from naturally occurring carboxylic acids and amines. These two comments should be deleted.

To some extent, the authors have made a scope that focuses on number of entries rather than diversity. It would be useful to have a bit more variety and testing/demonstrating the limitations. For example, for the α -diazo ketone scope in Figure 3b, five of the α -diazo ketones are used twice with different amides, which seems unnecessary. To probe diversity and limitations, the authors should test and include:

- An ortho-substituted α -diazo ketone.
- A stronger para-electron-donating group (MeO, not Me) on the α -diazo ketone.
- A secondary amide that is not used in competition with a primary amide. Will secondary amides react if there is no primary amide?
- An α -diazo ester.
- An α -diazo amide (can be secondary or tertiary amide).

The authors mention a meta-MeO as an electron-donating substituent, but in the meta-position the benzylic site will feel it as an electron-withdrawing group (as shown in the plot in Figure 5b). I believe the text accompanying Figure 3 should be rephrased to clarify this.

In the mechanistic section (lines 145-146), the authors state that they analyze the enantiodetermining “proton transfer step without the participation of the rhodium catalyst”. However, in the optimization table there is a clear effect on enantiomeric excess (ee) from the carboxylate on the rhodium catalyst when using the same squaramide (C8). For example, Rh2(piv)4 gives 76% ee, while Rh2(TPA)4 gives 90% ee. This difference equals to $\Delta\Delta G(\text{TS}) = 0.55$ kcal/mol, which is significant in comparison to the calculated $\Delta\Delta G(\text{TS})$ 1.3 kcal/mol. When there is such a large effect on ee from the rhodium catalyst, it is surprising that it is left out of the discussion about the enantiodetermining step. When there is a clear experimental effect, it is not enough to say that DFT ruled out that there should be an effect... because clearly there is an effect. An explanation for the effect of the rhodium complex on enantioselectivity should be included.

Given the basic nature of a pyridine (3z used in 100 times excess compared to C8), I am surprised that there is no effect on the reaction outcome, when all the enantiodetermining steps are proton transfers. If the authors have a hypothesis for this conundrum, it would be nice to include it as a note.

Comments for Supporting Information:

Many of the NMRs of products contain minor impurities, but I still think it is acceptable.

Nonetheless, the signal-to-noise ratio for ^{13}C NMR of 7e and 7h should be improved.

For the HPLC traces:

- The racemate of 3t and epimeric mixture of 5f should be repurified and cleaner HPLC traces made.
- For 3m, please integrate the major peak without the extra second peak.
- For the products from Grignard addition, the racemates on the HPLC traces are drawn as mixtures of both diastereomers and enantiomers, but there are only two peaks in the HPLC trace. Is the drawing supposed to show a racemic mixture of one diastereomer instead?
- The full HPLC trace should be used instead of truncated spectra. For example: 3h, 3m, 3aa, 4k, 5b, 7g.

Point to point response to reviewer's comments

Reviewer #1:

The manuscript submitted by Zhou, Liu and coworkers reported achiral rhodium and chiral squaramide co-catalyzed enantioselective carbene N–H insertion reaction, which provides a powerful tool for the synthesis of structural divergent chiral amides and related useful molecules. The authors have optimized the conditions to achieve high enantiomeric excesses for broad substrate scope under mild reaction conditions and at a rapid reaction rate. The reaction products were converted into valuable derivatives. Kinetic studies and computational studies were performed to elucidate the asymmetric proton transfer process in the reaction. Therefore, I would like to recommend the publication of this paper in Nat. Commun. if the authors can revise the manuscript appropriately. I would like to make some comments below.

1. Although rhodium catalysts work well in Table 1 and SI, have authors tried other metal catalysts besides $\text{Cu}(\text{MeCN})_4\text{PF}_6$ and CuTp^* ? I feel that many metal catalysts can catalyze this process, such as Pd, Fe catalysts etc.

Response: Thanks for the comment. We have tried several metal catalysts that have been verified to be capable of decomposing diazo compounds in others' works, but in our conditions they didn't show better reactivity than dirhodium acetate. The corresponding data has been added to the revised supporting information.

Entry ^a	x	[M]	Yield (%) ^b
1	1	$\text{Rh}_2(\text{OAc})_4$	80
2	5	$\text{Cu}(\text{MeCN})_4\text{PF}_6$	<5
3	5	CuTp^*	<5
4	5	$\text{Pd}(\text{MeCN})_2\text{Cl}_2$	trace
5	5	FeCl_3	trace
6	5	$[\text{Ir}(\text{COD})\text{Cl}]_2$	<5
7	5	CoBr_2	<5

^a Reaction conditions: α -diazoketone (0.2 mmol), 3-phenylpropionamide (0.2 mmol), DCM (2 mL), 20 °C, 12 h. ^b Isolated yield.

2. In DFT studies, the complexation of rhodium catalyst and squaramide(C=O) was ruled out by DFT calculations. Is it possible that chiral Rh complexes could be formed by the coordination interaction of $\text{Rh}_2(\text{TPA})_4$ with nitrogen atom of hydroquinine?

Response: Thanks for the suggestion. There are two possible coordination modes between rhodium and hydroquinine nitrogen atoms, that is, the coordination with quinuclidine N atom and the coordination with quinoline N atom. The first possibility will block the reactive site of the catalyst and was therefore ruled out. We calculated several transition states considering the second possibility, and found that energy barrier decreases 6.6 kcal/mol by coordination of rhodium with quinoline N atom, as shown in the figures below. Corresponding corrections has been made in the

revise manuscript and supporting information.

TS without Rh coordination:

TSR-I $\Delta G^\ddagger=12.6$ kcal/mol

TSS-I $\Delta G^\ddagger=13.9$ kcal/mol

TS with coordination of $\text{Rh}_2(\text{OAc})_4$:

RhTSRb-I $\Delta G^\ddagger=6.0$ kcal/mol

RhTSSb-I $\Delta G^\ddagger=6.5$ kcal/mol

TS with coordination of $\text{Rh}_2(\text{TPA})_4$ (energies are based on RhTPA-TSR):

RhTPA-TSR $\Delta G^\ddagger=0.0$ kcal/mol

RhTPA-TSS $\Delta G^\ddagger=2.2$ kcal/mol

3. It is interesting to know if this enantioselective carbene N–H bond insertion transformation is applicable to the alkyl-substituted diazo compounds.

Response: Thanks for the comment. We tried the insertion reaction using α -methyl diazo ketone, which turned out messy and no product obtained.

Reviewer #2:

Chiral amide bonds are ones of the most common in nature and their efficient synthesis remains challenging. Zhou and his co-workers report an efficient chiral amide synthetic methodology with a pretty rapid reaction rate under mild condition via the N-H insertion. The co-catalyst system works efficiently to afford an excellent enantioselective control. The reaction also gave a wide substrate scope and synthetic application in the chiral amide bond construction. I reaccommodate it to be published after the following questions are addressed:

1. It is seemed that the substrate, diazoketones, play important roles in the enantioselective control. As the authors illustrate in the reaction mechanism, the carbonyl group may serve as a hydrogen bond donor/acceptor in the key intermediates, then what will happen if remove the carbonyl group? and how about phenylacetate pyridine diazo? This should be mentioned and discussed in the mechanism investigation section.

Response: Thanks for the comments. We tried the insertion reaction using phenyl methyl diazomethane and 4-nitrophenyl phenyl diazomethane, which have no carbonyl group, but the results are messy and no product obtained. Reaction using phenyl diazo acetate was conducted and corresponding product was obtained with quantitative yield and 27% ee. This result shows the importance of the carbonyl group for the reaction. We also synthesized pyridyl-containing diazo compounds and run the insertion reactions. The diazo A could not be obtained by standard methods of preparing diazo compounds. The diazo B can react with butyramide to afford racemic product in 92% yield. The reaction of diazo C with butyramide gives messy result. A discussion on the function of carbonyl group in substrate has been added in the revised text and Figure 5c.

2. The proton shift is key in this reaction, so how water affects this reaction? Does it need dry solvent?

Response: Thanks for the comment. We tried the reaction with addition of 1 equivalent of water,

and obtained the product with lower yield and ee. It seems that dry solvent is needed to obtain better yield and enantiocontrol.

Entry	Additive	Yield (%)	ee (%)
1	--	99	92
2	1 equiv. H ₂ O	71	85

3. Do the chiral Rh catalysts work in this reaction?

Response: Thanks for the comment. We have tried several commonly used chiral Rh catalysts, all of which gave decent yield but no enantioselectivity.

Entry	Rh ₂ L ₄	Yield (%)	ee (%)
1	Rh ₂ (S-SPA) ₄	57	rac
2	Rh ₂ (R-BTPCP) ₄	63	rac
3	Rh ₂ (R-TCPTTL) ₄	59	rac
4	Rh ₂ (S-NTTL) ₄	62	rac
5	Rh ₂ (S-DOSP) ₄	46	rac

4. How about the ee value for each diastereomer of 5e-5h? These data need to be addressed.

Response: Thanks for the comment. Since the amide substrates we used for 5d-5h are optically pure, and theoretically no racemization of the amide substrates and epimerization of the amide side of the products can happen under mild reaction conditions, so each diastereomer of 5d-5h is also optically pure. Indeed, we did not find another enantiomer of each diastereomer of 5e-5h. We add ee value (>99%) to these products in Figure 3 of revised manuscript.

5. It is a little confused, the sentence (Line 111-112) "In the synthesis of chiral compounds, it is important to be able to selectively obtain either enantiomer." What the authors would like to declare?

Response: Thanks for the comment. In the sentence, we want to say that a good enantioselective synthesis method is one that can obtain either of the two enantiomers of chiral compound. We have rephrased this sentence more clearly in the revised manuscript.

6. Many language errors need to be corrected, for example:

Line 24, carbonyl should be carbonyl group

Line 64, the same ee value should be the same ee values

Line 82, Amide should be amides, alkyl should be alkyl group

Line 84 and 86, with high yields ... should be in high yields and with high enantioselectivities...

Line 87, "The amidesalso underwent the reaction" should be "The reaction using amides.....also underwent well."

Line 88, with high yields should be in high yields

Line 105, showed should be show

Line 106, demonstrated should be demonstrate; "to" should be "for"

Line 120, with excellent yield should in excellent yield

Line 130, decomposes should be is decomposed

Line 133, release should be releases

Response: Thanks for pointing out the errors. These errors have been corrected in the revised manuscript.

Reviewer #3:

In this manuscript, Zhou et al. report an enantioselective carbene insertion into amide N-H bonds catalyzed by an achiral rhodium complex and a chiral squaramide. The catalyst loadings are low, the reaction times short, and both yields and enantioselectivities high. The scope is good, but more diversity and information about limitations are warranted. DFT is used to explain the enantiocontrol enforced by the squaramide catalyst, but the observed experimental effect of the rhodium catalyst on enantioselectivity is not discussed. Although enantioselective carbene insertion into N-H bonds have been reported before, the amides in this paper are significantly different from previous nitrogen coupling partners. As such, I would evaluate the novelty to be suitable for Nature Communication.

Overall, there are some issues to address; nonetheless, the manuscript could become suitable for Nature Communication after addressing the comments outlined below.

Two places in the introduction, I find the phrasing unfair: "...reduction of hazardous by-products..." and "...N₂ as the sole environmentally friendly by-product". The α -diazo ketone is not a naturally occurring feedstock material; it must be synthesized. As such the α -diazo ketone synthesis should be included in the consideration of the environmental impact, especially when the authors are comparing it to amide bond formation from naturally occurring carboxylic acids and amines. These two comments should be deleted.

Response: Thanks for the suggestion. These two phrases have been deleted in the revised manuscript.

To some extent, the authors have made a scope that focuses on number of entries rather than diversity. It would be useful to have a bit more variety and testing/demonstrating the limitations. For example, for the α -diazo ketone scope in Figure 3b, five of the α -diazo ketones are used twice with different amides, which seems unnecessary. To probe diversity and limitations, the authors should test and include:

- An ortho-substituted α -diazo ketone.

Response: Thanks for the suggestion. We run an insertion reaction using ortho-F-substituted α -diazo ketone. The result has been added to the revised manuscript and SI.

- A stronger para-electron-donating group (MeO, not Me) on the α -diazo ketone.

Response: Thanks for the suggestion. We have tried several methods to synthesize para-MeO-substituted phenyl α -diazo ketone, but it is not stable and can not be obtained. So is para-Me₂N-substituted phenyl α -diazo ketone.

- A secondary amide that is not used in competition with a primary amide. Will secondary amides react if there is no primary amide?

Response: Thanks for the suggestion. We run the insertion reactions using N-Me benzamide, N-acyl aniline and 2-piperidone. However, no product was detected and the reactions turned messy.

- An α -diazo ester.

Response: Thanks for the suggestion. We run the insertion reaction using methyl α -diazophenylacetate and obtained the insertion product with quantitative yield and low ee. This result has been mentioned in the revised manuscript.

- An α -diazo amide (can be secondary or tertiary amide).

Response: Thanks for the suggestion. We run the insertion reaction using α -diazophenylacetamide and obtained the insertion product with moderate yield and low ee. This result has been mentioned in the revised manuscript.

0

57% yield, 25% ee

The authors mention a meta-MeO as an electron-donating substituent, but in the meta-position the benzylic site will feel it as an electron-withdrawing group (as shown in the plot in Figure 5b). I believe the text accompanying Figure 3 should be rephrased to clarify this.

Response: Thanks for the comment. The text and Figure 3 have been rephrased in the revised manuscript.

In the mechanistic section (lines 145-146), the authors state that they analyze the enantiodetermining “proton transfer step without the participation of the rhodium catalyst”. However, in the optimization table there is a clear effect on enantiomeric excess (ee) from the carboxylate on the rhodium catalyst when using the same squaramide (C8). For example, $\text{Rh}_2(\text{piv})_4$ gives 76% ee, while $\text{Rh}_2(\text{TPA})_4$ gives 90% ee. This difference equals to $\Delta\Delta G(\text{TS}) = 0.55$ kcal/mol, which is significant in comparison to the calculated $\Delta\Delta G(\text{TS})$ 1.3 kcal/mol. When there is such a large effect on ee from the rhodium catalyst, it is surprising that it is left out of the discussion about the enantiodetermining step. When there is a clear experimental effect, it is not enough to say that DFT ruled out that there should be an effect... because clearly there is an effect. An explanation for the effect of the rhodium

complex on enantioselectivity should be included.

Response: Thanks for the comment. We calculated several transition states which consider the coordination between rhodium with squaramide catalyst and indeed found that rhodium complex can coordinate with the squaramide catalyst during the proton transfer step and thereby can influence the enantiocontrol. Calculations show that coordination of $\text{Rh}_2(\text{OAc})_4$ with the oxygen of the squaramide catalyst doesn't have obvious effect on energy (RhTSRa-I). However, when $\text{Rh}_2(\text{OAc})_4$ is coordinated with the quinoline group of the squaramide catalyst, the energy barrier is significantly reduced by 6.6 kcal/mol (RhTSRb-I). Meanwhile, the change of rhodium complex in the calculation model from $\text{Rh}_2(\text{OAc})_4$ to $\text{Rh}_2(\text{TPA})_4$ lead to an increase of $\Delta\Delta G^\ddagger$ between the transition states of the major product and the minor product (2.2 kcal/mol, RhTPA-TSS vs RhTPA-TSR), which can explain why different rhodium complex gives different ee value. Corresponding DFT calculation data and discussion have been added in the revised manuscript and supporting information.

TS without Rh coordination (energies are based on free enol):

TS with coordination of $\text{Rh}_2(\text{OAc})_4$ with oxygen (energies are based on free enol):

TS with coordination of $\text{Rh}_2(\text{OAc})_4$ with quinoline (energies are based on free enol):

TS with coordination of $\text{Rh}_2(\text{TPA})_4$ (energies are based on RhTPA-TSR):

RhTPA-TSR $\Delta G^\ddagger=0.0$ kcal/mol

RhTPA-TSS $\Delta G^\ddagger=2.2$ kcal/mol

Given the basic nature of a pyridine (3z used in 100 times excess compared to C8), I am surprised that there is no effect on the reaction outcome, when all the enantiodetermining steps are proton transfers. If the authors have a hypothesis for this conundrum, it would be nice to include it as a note.

Response: Thanks for the suggestion. The pK_a (DMSO) of the conjugate acid of quinuclidine is 9.8, which is 6 orders of magnitude larger than the conjugate acid of pyridine (3.4). (ref: Eur. J. Org. Chem. 2019, 22, 6735–6748.) So, even if 3z is 100 times more than C8, the pyridyl group will not interfere with the reaction. Corresponding explanation has been added in the revised manuscript.

Comments for Supporting Information:

Many of the NMRs of products contain minor impurities, but I still think it is acceptable. Nonetheless, the signal-to-noise ratio for ^{13}C NMR of 7e and 7h should be improved.

Response: The products **7e** and **7h** have been re-purified and taken ^{13}C NMR analysis. The improved spectra have been added in the revised supporting information.

For the HPLC traces:

- The racemate of 3t and epimeric mixture of 5f should be re-purified and cleaner HPLC traces made.

Response: These samples have been re-purified and the improved HPLC charts have been added in the revised supporting information.

- For 3m, please integrate the major peak without the extra second peak.

Response: The sample of 3m has been re-purified and taken HPLC analysis. The HPLC trace and data have been corrected in the revised supporting information and manuscript.

- For the products from Grignard addition, the racemates on the HPLC traces are drawn as mixtures of both diastereomers and enantiomers, but there are only two peaks in the HPLC trace. Is the drawing supposed to show a racemic mixture of one diastereomer instead?

Response: Racemate HPLC traces show only a racemic mixture of one diastereomer. Due to the very high diastereoselectivity of the Grignard addition reaction ($dr >20:1$), another diastereoisomer cannot be seen in NMR. These have been clarified in the HPLC charts of Grignard addition products in the revised supporting information.

- The full HPLC trace should be used instead of truncated spectra. For example: 3h, 3m, 3aa, 4k, 5b, 7g.

Response: Full HPLC traces of these samples have been used in the revised supporting information.

REVIEWERS' COMMENTS

Reviewer #1 (Remarks to the Author):

The manuscript submitted by Zhou, Liu and coworkers has been revised based on the comments of the reviewers. My concerns have been addressed. The authors have made the corresponding changes, and this manuscript is ready for publication.

Reviewer #2 (Remarks to the Author):

All the questions have been well addressed, so I think this revised manuscript is acceptable to be published on Nature Communications.

Reviewer #3 (Remarks to the Author):

The authors have comprehensively addressed all the comments from the reviewers. I therefore recommend that this paper is accepted.